# Differentiable Quality Diversity

**Matthew C. Fontaine**
University of Southern California
Los Angeles, CA
mfontain@usc.edu

**Stefanos Nikolaidis**
University of Southern California
Los Angeles, CA
nikolaid@usc.edu

## Abstract

Quality diversity (QD) is a growing branch of stochastic optimization research that studies the problem of generating an archive of solutions that maximize a given objective function but are also diverse with respect to a set of specified measure functions. However, even when these functions are differentiable, QD algorithms treat them as "black boxes", ignoring gradient information. We present the differentiable quality diversity (DQD) problem, a special case of QD, where both the objective and measure functions are first order differentiable. We then present MAP-Elites via a Gradient Arborescence (MEGA), a DQD algorithm that leverages gradient information to efficiently explore the joint range of the objective and measure functions. Results in two QD benchmark domains and in searching the latent space of a StyleGAN show that MEGA significantly outperforms state-of-the-art QD algorithms, highlighting DQD's promise for efficient quality diversity optimization when gradient information is available. Source code is available at https://github.com/icaros-usc/dqd.

## 1 Introduction

We introduce the problem of differentiable quality diversity (DQD) and propose the MAP-Elites via a Gradient Arborescence (MEGA) algorithm as the first DQD algorithm.

Unlike single-objective optimization, quality diversity (QD) is the problem of finding a range of high quality solutions that are diverse with respect to prespecified metrics. For example, consider the problem of generating realistic images that match as closely as possible a target text prompt "Elon Musk", but vary with respect to hair and eye color. We can formulate the problem of searching the latent space of a generative adversarial network (GAN) [26] as a QD problem of discovering latent codes that generate images maximizing a matching score for the prompt "Elon Musk", while achieving a diverse range of measures of hair and eye color, assessed by visual classification models [51]. More generally, the QD objective is to maximize an objective $f$ for each output combination of measure functions $m_i$. A QD algorithm produces an archive of solutions, where the algorithm attempts to discover a representative for each measure output combination, whose $f$ value is as large as possible.

While our example problem can be formulated as a QD problem, all current QD algorithms treat the objective $f$ and measure functions $m_i$ as a black box. This means, in our example problem, current QD algorithms fail to take advantage of the fact that both $f$ and $m_i$ are end-to-end differentiable neural networks. Our proposed differentiable quality diversity (DQD) algorithms leverage first-order derivative information to significantly improve the computational efficiency of solving a variety of QD problems where $f$ and $m_i$ are differentiable.

To solve DQD, we introduce the concept of a *gradient arborescence*. Like gradient ascent, a gradient arborescence makes greedy ascending steps based on the objective $f$. Unlike gradient ascent, a gradient arborescence encourages exploration by branching via the measures $m_i$. We adopt the term

35th Conference on Neural Information Processing Systems (NeurIPS 2021).

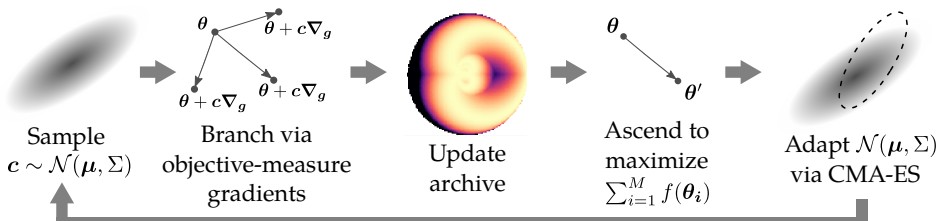

Figure 1: An overview of the Covariance Matrix Adaptation MAP-Elites via a Gradient Arborescence (CMA-MEGA) algorithm. The algorithm leverages a gradient arborescence to branch in objective-measure space, while dynamically adapting the gradient steps to maximize a QD objective (Eq. 1).

*arborescence* from the minimum arborescence problem [8] in graph theory, a directed counterpart to the minimum spanning tree problem, to emphasize the directedness of the branching search.

Our work makes four main contributions. 1) We introduce and formalize the problem of differentiable quality diversity (DQD). 2) We propose two DQD algorithms: Objective and Measure Gradient MAP-Elites via a Gradient Arborescence (OMG-MEGA), an algorithm based on MAP-Elites [13], which branches based on the measures $m_i$ but ascends based on the objective function $f$; and Covariance Matrix Adaptation MEGA (CMA-MEGA) which is based on the CMA-ME [19] algorithm, and which branches based on the objective-measure space but ascends based on maximizing the QD objective (Fig. 1). Both algorithms search directly in measure space and leverage the gradients of $f$ and $m_i$ to form efficient parameter space steps in $\boldsymbol{\theta}$. 3) We show in three different QD domains (the linear projection, the arm repertoire, and the latent space illumination (LSI) domains), that DQD algorithms significantly outperform state-of-the-art QD algorithms that treat the objective and measure functions as a black box. 4) We demonstrate how searching the latent space of a StyleGAN [36] in the LSI domain with CMA-MEGA results in a diverse range of high-quality images.

## 2    Problem Definition

**Quality Diversity.** The quality diversity (QD) problem assumes an objective $f : \mathbb{R}^n \to \mathbb{R}$ in an $n$-dimensional continuous space $\mathbb{R}^n$ and $k$ measures $m_i : \mathbb{R}^n \to \mathbb{R}$ or, as a joint measure, $\boldsymbol{m} : \mathbb{R}^n \to \mathbb{R}^k$. Let $S = \boldsymbol{m}(\mathbb{R}^n)$ be the measure space formed by the range of $m$. For each $\boldsymbol{s} \in S$ the QD objective is to find a solution $\boldsymbol{\theta} \in \mathbb{R}^n$ such that $\boldsymbol{m}(\boldsymbol{\theta}) = \boldsymbol{s}$ and $f(\boldsymbol{\theta})$ is maximized.

However, we note that $\mathbb{R}^k$ is continuous, and an algorithm solving the quality diversity problem would require infinite memory to store all solutions. Thus, QD algorithms in the MAP-Elites [43, 13] family approximate the problem by discretizing $S$ via a tessellation method. Let $T$ be the tessellation of $S$ into $M$ cells. We relax the QD objective to find a set of solutions $\boldsymbol{\theta_i}, i \in \{1, \ldots, M\}$, such that each $\boldsymbol{\theta_i}$ occupies one unique cell in $T$. The occupants $\boldsymbol{\theta_i}$ of all $M$ cells form an archive of solutions. Each solution $\boldsymbol{\theta_i}$ has a position in the archive $\boldsymbol{m}(\boldsymbol{\theta_i})$, corresponding to one out of $M$ cells, and an objective value $f(\boldsymbol{\theta_i})$.

The objective of QD can be rewritten as follows, where the goal is to maximize the objective value for each cell in the archive:

$$\max \sum_{i=1}^{M} f(\boldsymbol{\theta_i}) \tag{1}$$

**Differentiable Quality Diversity.** We define the differentiable quality diversity (DQD) problem, as a QD problem where both the objective $f$ and measures $m_i$ are *first-order differentiable*.

## 3    Preliminaries

We present several state-of-the-art derivative-free QD algorithms. Our proposed DQD algorithm MEGA builds upon ideas from these works, while introducing measure and objective gradients into the optimization process.

**MAP-Elites and MAP-Elites (line)**. MAP-Elites [13, 43] first tessellates the measure space $S$ into evenly-spaced grid cells. The upper and lower bounds for $\boldsymbol{m}$ are given as input to constrain $S$ to a finite region. MAP-Elites first samples solutions from a fixed distribution $\boldsymbol{\theta} \sim \mathcal{N}(\mathbf{0}, I)$, and populates an initial archive after computing $f(\boldsymbol{\theta})$ and $\boldsymbol{m}(\boldsymbol{\theta})$. Each iteration of MAP-Elites selects $\lambda$ cells uniformly at random from the archive and perturbs each occupant $\boldsymbol{\theta_i}$ with fixed-variance $\sigma$ isotropic Gaussian noise: $\boldsymbol{\theta'} = \boldsymbol{\theta_i} + \sigma \mathcal{N}(\mathbf{0}, I)$. Each new candidate solution $\boldsymbol{\theta'}$ is then evaluated and added to the archive if $\boldsymbol{\theta'}$ discovers a new cell or improves an existing cell. The algorithm continues to generate solutions for a specified number of iterations.

Later work introduced the Iso+LineDD operator [61]. The Iso+LineDD operator samples two archive solutions $\boldsymbol{\theta_i}$ and $\boldsymbol{\theta_j}$, then blends a Gaussian perturbation with a noisy interpolation given hyperparameters $\sigma_1$ and $\sigma_2$: $\boldsymbol{\theta'} = \boldsymbol{\theta_i} + \sigma_1 \mathcal{N}(\mathbf{0}, I) + \sigma_2 \mathcal{N}(0, 1)(\boldsymbol{\theta_i} - \boldsymbol{\theta_j})$. In this paper we refer to MAP-Elites with an Iso+LineDD operator as MAP-Elites (line).

**CMA-ME**. Covariance Matrix Adaptation MAP-Elites (CMA-ME) [19] combines the archiving mechanisms of MAP-Elites with the adaptation mechanisms of CMA-ES [30]. While MAP-Elites creates new solutions by perturbing existing solutions with fixed-variance Gaussian noise, CMA-ME maintains a full-rank Gaussian distribution $\mathcal{N}(\boldsymbol{\mu}, \Sigma)$ in parameter space $\mathbb{R}^n$. Each iteration of CMA-ME samples $\lambda$ candidate solutions $\boldsymbol{\theta_i} \sim \mathcal{N}(\boldsymbol{\mu}, \Sigma)$, evaluates each solution, and updates the archive based on the following rule: if there is a previous occupant $\boldsymbol{\theta_p}$ at the same cell, we compute $\Delta_i = f(\boldsymbol{\theta_i}) - f(\boldsymbol{\theta_p})$, otherwise if the cell is empty we compute $\Delta_i = f(\boldsymbol{\theta_i})$. We then rank the sampled solutions by increasing improvement $\Delta_i$, with an extra criteria that candidates discovering new cells are ranked higher than candidates that improve existing cells. We then update $\mathcal{N}(\boldsymbol{\mu}, \Sigma)$ with the standard CMA-ES update rules based on the improvement ranking. CMA-ME restarts when all $\lambda$ solutions fail to change the archive. On a restart we reset the Gaussian $\mathcal{N}(\boldsymbol{\theta_i}, I)$, where $\boldsymbol{\theta_i}$ is an archive solution chosen uniformly at random, and all internal CMA-ES parameters. In the supplemental material, we derive, for the first time, a natural gradient interpretation of CMA-ME's improvement ranking mechanism, based on previous theoretical work on CMA-ES [1].

## 4  Algorithms

We present two variants of our proposed MEGA algorithm: OMG-MEGA and CMA-MEGA. We form each variant by adapting the concept of a gradient arborescence to the MAP-Elites and CMA-ME algorithms, respectively. Finally, we introduce two additional baseline algorithms, OG-MAP-Elites and OG-MAP-Elites (line), which operate only on the gradients of the objective.

**OMG-MEGA.** We first derive the Objective and Measure Gradient MAP-Elites via Gradient Arborescence (OMG-MEGA) algorithm from MAP-Elites.

First, we observe how gradient information could benefit a QD algorithm. Note that the QD objective is to explore the measure space, while maximizing the objective function $f$. We observe that maximizing a linear combination of measures : $\sum_{j=1}^{k} c_j m_j(\boldsymbol{\theta})$, where $\boldsymbol{c}$ is a $k$-dimensional vector of coefficients, enables movement in a $k$-dimensional measure space. Adding the objective function $f$ to the linear sum enables movement in an objective-measure space. Maximizing $g$ with a positive coefficient of $f$ results in steps that increasing $f$, while the direction of movement in the measure space is determined by the sign and magnitude of the coefficients $c_j$.

$$g(\boldsymbol{\theta}) = |c_0| f(\boldsymbol{\theta}) + \sum_{j=1}^{k} c_j m_j(\boldsymbol{\theta}) \tag{2}$$

We can then derive a direction function that perturbs a given solution $\boldsymbol{\theta}$ based on the gradient of our linear combination $g$: $\boldsymbol{\nabla} g(\boldsymbol{\theta}) = |c_0| \boldsymbol{\nabla} f(\boldsymbol{\theta}) + \sum_{j=1}^{k} c_j \boldsymbol{\nabla} m_j(\boldsymbol{\theta})$ . We incorporate the direction function $\boldsymbol{\nabla} g$ to derive a gradient-based MAP-Elites variation operator.

We observe that MAP-Elites samples a cell $\boldsymbol{\theta_i}$ and perturbs the occupant with Gaussian noise: $\boldsymbol{\theta'} = \boldsymbol{\theta_i} + \sigma\mathcal{N}(\mathbf{0}, I)$. Instead, we sample coefficents $\boldsymbol{c} \sim \mathcal{N}(\mathbf{0}, \sigma_g I)$ and step:

$$\boldsymbol{\theta'} = \boldsymbol{\theta_i} + \mid c_0 \mid \boldsymbol{\nabla}f(\boldsymbol{\theta_i}) + \sum_{j=1}^{k} c_j \boldsymbol{\nabla}m_j(\boldsymbol{\theta_i}) \tag{3}$$

The value $\sigma_g$ acts as a learning rate for the gradient step, because it controls the scale of the coefficients $c \sim \mathcal{N}(0, \sigma_g I)$. To balance the contribution of each function, we normalize all gradients. In the supplemental material, we further justify gradient normalization and provide an empirical ablation study. Other than our new gradient-based operator, OMG-MEGA is identical to MAP-Elites.

**CMA-MEGA.** Next, we derive the Covariance Matrix Adaptation MAP-Elites via a Gradient Arborescence (CMA-MEGA) algorithm from CMA-ME. Fig. 1 shows an overview of the algorithm.

First, we note that we sample $\boldsymbol{c}$ in OMG-MEGA from a fixed-variance Gaussian. However, it would be beneficial to select $\boldsymbol{c}$ based on how $\boldsymbol{c}$, and the subsequent gradient step on $\boldsymbol{\theta}$, improve the QD objective defined in equation 1.

We frame the selection of $\boldsymbol{c}$ as an optimization problem with the objective of maximizing the QD objective (Eq. 1). We model a distribution of coefficients $\boldsymbol{c}$ as a $k+1$-dimensional Gaussian $\mathcal{N}(\boldsymbol{\mu}, \Sigma)$. Given a $\boldsymbol{\theta}$, we can sample $\boldsymbol{c} \sim \mathcal{N}(\boldsymbol{\mu}, \Sigma)$, compute $\boldsymbol{\theta'}$ via Eq. 3, and adapt $\mathcal{N}(\boldsymbol{\mu}, \Sigma)$ towards the direction of maximum increase of the QD objective (see Eq. 1).

We follow an evolution strategy approach to model and dynamically adapt the sampling distribution of coefficients $\mathcal{N}(\boldsymbol{\mu}, \Sigma)$. We sample a population of $\lambda$ coefficients from $\boldsymbol{c_i} \sim \mathcal{N}(\boldsymbol{\mu}, \Sigma)$ and generate $\lambda$ solutions $\boldsymbol{\theta_i}$. We then compute $\Delta_i$ from CMA-ME's improvement ranking for each candidate solution $\boldsymbol{\theta_i}$. By updating $\mathcal{N}(\boldsymbol{\mu}, \Sigma)$ with CMA-ES update rules for the ranking $\Delta_i$, we dynamically adapt the distribution of coefficients $\boldsymbol{c}$ to maximize the QD objective.

Algorithm 1 shows the pseudocode for CMA-MEGA. In line 3 we evaluate the current solution and compute an objective value $f$, a vector of measure values $\boldsymbol{m}$, and gradient vectors. As we dynamically adapt the coefficients $\boldsymbol{c}$, we normalize the objective and measure gradients (line 4) for stability. Because the measure space is tessellated, the measures $\boldsymbol{m}$ place solution $\boldsymbol{\theta}$ into one of the $M$ cells in the archive. We then add the solution to the archive (line 5), if the solution discovers an empty cell in the archive, or if it improves an existing cell, identically to MAP-Elites.

We then use the gradient information to compute a step that maximizes improvement of the archive. In lines 6-12, we sample a population of $\lambda$ coefficients from a multi-variate Gaussian retained by CMA-ES, and take a gradient step for each sample. We evaluate each sampled solution $\boldsymbol{\theta'_i}$, and compute the improvement $\Delta_i$ (line 11). As in CMA-ME, we specify $\Delta_i$ as the difference in the objective value between the sampled solution $\boldsymbol{\theta_i}$ and the existing solution, if one exists, or as the absolute objective value of the sampled solution if $\boldsymbol{\theta_i}$ belongs to an empty cell.

In line 13, we rank the sampled gradients $\boldsymbol{\nabla}_i$ based on their respective improvements. As in CMA-ME, we prioritize exploration of the archive by ranking first by their objective values all samples that discover new cells, and subsequently all samples that improve existing cells by their difference in improvement. We then compute an ascending gradient step as a linear combination of gradients (line 14), following the recombination weights $w_i$ from CMA-ES [30] based on the computed improvement ranking. These weights correspond to the log-likelihood probabilities of the samples in the natural gradient interpretation of CMA-ES [1].

In line 16, CMA-ES adapts the multi-variate Gaussian $\mathcal{N}(\boldsymbol{\mu}, \Sigma)$, as well as internal search parameters $\boldsymbol{p}$, from the improvement ranking of the coefficients. In the supplemental material, we provide a natural gradient interpretation of the improvement ranking rules of CMA-MEGA, where we show that the coefficient distribution of CMA-MEGA approximates natural gradient steps of maximizing a modified QD objective.

**CMA-MEGA (Adam).** We add an Adam-based variant of CMA-MEGA, where we replace line 15 with an Adam gradient optimization step [38].

**OG-MAP-Elites.** To show the importance of taking gradient steps in the measure space, as opposed to only taking gradient steps in objective space and directly perturbing the parameters, we derive two variants of MAP-Elites as a baseline that draw insights from the recently proposed Policy Gradient

---

**Algorithm 1** Covariance Matrix Adaptation MAP-Elites via a Gradient Aborescence (CMA-MEGA)

---

**CMA-MEGA** $(evaluate, \boldsymbol{\theta_0}, N, \lambda, \eta, \sigma_g)$

> **input :** An evaluation function $evaluate$ which computes the objective, the measures, gradients of the objective and measures, an initial solution $\boldsymbol{\theta_0}$, a desired number of iterations $N$, a branching population size $\lambda$, a learning rate $\eta$, and an initial step size for CMA-ES $\sigma_g$.
> **result :** Generate $N(\lambda + 1)$ solutions storing elites in an archive $A$.

1   Initialize solution parameters $\boldsymbol{\theta}$ to $\boldsymbol{\theta_0}$, CMA-ES parameters $\boldsymbol{\mu} = \boldsymbol{0}$, $\Sigma = \sigma_g I$, and $\boldsymbol{p}$, where we let $\boldsymbol{p}$ be the CMA-ES internal parameters.

2   **for** $iter \leftarrow 1$ **to** $N$ **do**

3      $f, \boldsymbol{\nabla}_f, \boldsymbol{m}, \boldsymbol{\nabla_m} \leftarrow \text{evaluate}(\boldsymbol{\theta})$

4      $\boldsymbol{\nabla}_f \leftarrow \text{normalize}(\boldsymbol{\nabla}_f), \boldsymbol{\nabla}_m \leftarrow \text{normalize}(\boldsymbol{\nabla}_m)$

5      $\text{update\_archive}(\boldsymbol{\theta}, f, \boldsymbol{m})$

6      **for** $i \leftarrow 1$ **to** $\lambda$ **do**

7          $\boldsymbol{c} \sim \mathcal{N}(\boldsymbol{\mu}, \Sigma)$

8          $\boldsymbol{\nabla}_i \leftarrow c_0 \boldsymbol{\nabla}_f + \sum_{j=1}^{k} c_j \boldsymbol{\nabla}_{m_j}$

9          $\boldsymbol{\theta'_i} \leftarrow \boldsymbol{\theta} + \boldsymbol{\nabla}_i$

10         $f', *, \boldsymbol{m'}, * \leftarrow \text{evaluate}(\boldsymbol{\theta'_i})$

11         $\Delta_i \leftarrow \text{update\_archive}(\boldsymbol{\theta'_i}, f', \boldsymbol{m'})$

12      **end**

13      rank $\boldsymbol{\nabla}_i$ by $\Delta_i$

14      $\boldsymbol{\nabla}_{\text{step}} \leftarrow \sum_{i=1}^{\lambda} w_i \boldsymbol{\nabla}_{\text{rank[i]}}$

15      $\boldsymbol{\theta} \leftarrow \boldsymbol{\theta} + \eta \boldsymbol{\nabla}_{\text{step}}$

16      Adapt CMA-ES parameters $\boldsymbol{\mu}, \Sigma, \boldsymbol{p}$ based on improvement ranking $\Delta_i$

17      **if** *there is no change in the archive* **then**

18         Restart CMA-ES with $\boldsymbol{\mu} = 0, \Sigma = \sigma_g I$.

19         Set $\boldsymbol{\theta}$ to a randomly selected existing cell $\boldsymbol{\theta_i}$ from the archive

20      **end**

21   **end**

---

Assisted MAP-Elites (PGA-ME) algorithm [45]. PGA-ME combines the Iso+LineDD operator [61] with a policy gradient operator only on the objective. Similarly, our proposed Objective-Gradient MAP-Elites (OG-MAP-Elites) algorithm combines an objective gradient step with a MAP-Elites style perturbation operator. Each iteration of OG-MAP-Elites samples $\lambda$ solutions $\boldsymbol{\theta_i}$ from the archive. Each $\boldsymbol{\theta_i}$ is perturbed with Gaussian noise to form a new candidate solution $\boldsymbol{\theta'_i} = \boldsymbol{\theta_i} + \sigma \mathcal{N}(\boldsymbol{0}, I)$. OG-MAP-Elites evaluates the solution and updates the archive, exactly as in MAP-Elites. However, OG-MAP-Elites takes one additional step: for each $\boldsymbol{\theta'_i}$, the algorithm computes $\boldsymbol{\nabla} f(\boldsymbol{\theta'_i})$, forms a new solution $\boldsymbol{\theta''_i} = \boldsymbol{\theta'_i} + \eta \boldsymbol{\nabla} f(\boldsymbol{\theta'_i})$ with an objective gradient step, and evaluates $\boldsymbol{\theta''_i}$. Finally, we update the archive with all solutions $\boldsymbol{\theta'_i}$ and $\boldsymbol{\theta''_i}$.

**OG-MAP-Elites (line).** Our second baseline, OG-MAP-Elites (line) replaces the Gaussian operator with the Iso+LineDD operator [61]: $\boldsymbol{\theta'} = \boldsymbol{\theta_i} + \sigma_1 \mathcal{N}(\boldsymbol{0}, I) + \sigma_2 \mathcal{N}(\boldsymbol{0}, 1)(\boldsymbol{\theta_i} - \boldsymbol{\theta_j})$. We consider OG-MAP-Elites (line) a DQD variant of PGA-ME. However, PGA-ME was designed as a reinforcement learning (RL) algorithm, thus many of the advantages gained in RL settings are lost in OG-MAP-Elites (line). We provide a detailed discussion and ablations in the supplemental material.

## 5 Domains

DQD requires differentiable objective and measures, thus we select benchmark domains from previous work in the QD literature where we can compute the gradients of the objective and measure functions.

**Linear Projection.** To show the importance of adaptation mechanisms in QD, the CMA-ME paper [19] introduced a simple domain, where reaching the extremes of the measures is challenging for non-adaptive QD algorithms. The domain forms each measure $m_i$ by a linear projection from $\mathbb{R}^n$ to $\mathbb{R}$, while bounding the contribution of each component $\boldsymbol{\theta}_i$ to the range $[-5.12, 5.12]$.

We note that uniformly sampling from a hypercube in $\mathbb{R}^n$ results in a narrow distribution of the linear projection in $\mathbb{R}$ [19, 34]. Increasing the number of parameters $n$ makes the problem of covering the

| Algorithm | LP (sphere) | | LP (Rastrigin) | | Arm Repertoire | | LSI | |
|---|---|---|---|---|---|---|---|---|
| | QD-score | Coverage | QD-score | Coverage | QD-score | Coverage | QD-score | Coverage |
| MAP-Elites | 1.04 | 1.17% | 1.18 | 1.72% | 1.97 | 8.06% | 13.88 | 23.15% |
| MAP-Elites (line) | 12.21 | 14.32% | 8.12 | 11.79% | 33.51 | 35.79% | 16.54 | 25.73% |
| CMA-ME | 1.08 | 1.21% | 1.21 | 1.76% | 55.98 | 56.95% | 18.96 | 26.18% |
| OG-MAP-Elites | 1.52 | 1.67% | 0.83 | 1.26% | 57.17 | 58.08% | N/A | N/A |
| OG-MAP-Elites (line) | 15.01 | 17.41% | 6.10 | 8.85% | 59.66 | 60.28% | N/A | N/A |
| OMG-MEGA | 71.58 | 92.09% | 55.90 | 77.00% | 44.12 | 44.13% | N/A | N/A |
| CMA-MEGA | 75.29 | **100.00%** | 62.54 | **100.00%** | **74.18** | **74.18%** | 5.36 | 8.61% |
| CMA-MEGA (Adam) | **75.30** | **100.00%** | **62.58** | **100.00%** | 73.82 | 73.82% | **21.82** | **30.73%** |

Table 1: Mean QD-score and coverage values after 10,000 iterations for each algorithm per domain.

measure space more challenging, because to reach an extremum $m_i(\boldsymbol{\theta}) = \pm 5.12n$, all components must equal the extremum: $\boldsymbol{\theta}[i] = \pm 5.12$.

We select this domain as a benchmark to highlight the need for adaptive gradient coefficients for CMA-MEGA as opposed to constant coefficients for OMG-MEGA, because reaching the edges of the measure space requires dynamically shrinking the gradient steps.

As a QD domain, the domain must provide an objective. The CMA-ME study [19] introduces two variants of the linear projection domain with an objective based on the sphere and Rastrigin functions from the continuous black-box optimization set of benchmarks [29, 31]. We optimize an $n = 1000$ unbounded parameter space $\mathbb{R}^n$. We provide more detail in the supplemental material.

**Arm Repertoire.** We select the robotic arm repertoire domain from previous work [13, 61]. The goal in this domain is to find an inverse kinematics (IK) solution for each reachable position of the end-effector of a planar robotic arm with revolute joints. The objective $f$ of each solution is to minimize the variance of the joint angles, while the measure functions are the positions of the end effector in the $x$ and $y$-axis, computed with the forward kinematics of the planar arm [44]. We selected a 1000-DOF robotic arm.

**Latent Space Illumination.** Previous work [20] introduced the problem of exploring the latent space of a generative model directly with a QD algorithm. The authors named the problem latent space illumination (LSI). As the original LSI work evaluated non-differentiable objectives and measures, we create a new benchmark for the differentiable LSI problem by generating images with StyleGAN [36] and leveraging CLIP [51] to create differentiable objective and measure functions. We adopt the StyleGAN+CLIP [48] pipeline, where StyleGAN-generated images are passed to CLIP, which in turn evaluates how well the generated image matches a given text prompt. We form the prompt "Elon Musk with short hair." as the objective and for the measures we form the prompts "A person with red hair." and "A man with blue eyes.". The goal of DQD becomes generating faces similar to Elon Musk with short hair, but varying with respect to hair and eye color.

## 6 Experiments

We conduct experiments to assess the performance of the MEGA variants. In addition to our OG-MAP-Elites baselines, which we propose in section 4, we compare the MEGA variants with the state-of-the-art QD algorithms presented in section 3. We implemented MEGA and OG-MAP-Elites variants in the Pyribs [59] QD library and compare against the existing Pyribs implementations of MAP-Elites, MAP-Elites (line), and CMA-ME.

### 6.1 Experiment Design

**Independent Variables.** We follow a between-groups design, where the independent variables are the algorithm and the domain (linear projection, arm repertoire, and LSI). We did not run OMG-MEGA and OG-MAP-Elites in the LSI domain; while CMA-MEGA computes the $f$ and $m_i$ gradients once per iteration (line 3 in Algorithm 1), OMG-MEGA and OG-MAP-Elites compute the $f$ and $m_i$ gradients for every sampled solution, making their execution cost-prohibitive for the LSI domain.

**Dependent Variables.** We measure both the diversity and the quality of the solutions returned by each algorithm. These are combined by the QD-score metric [49], which is defined as the sum of $f$ values of all cells in the archive (Eq. 1). To make the QD-score invariant with respect to the

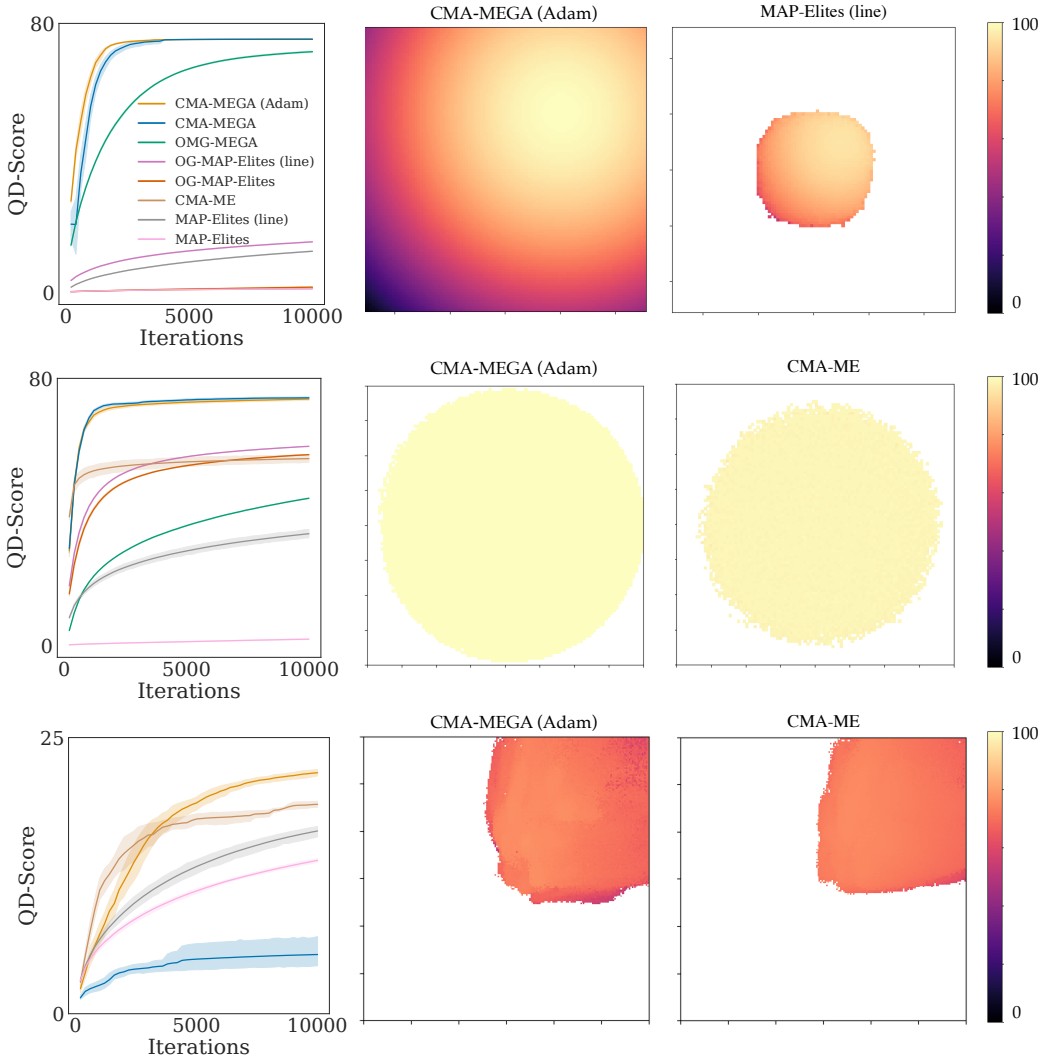

Figure 2: QD-Score plot with 95% confidence intervals and heatmaps of generated archives by CMA-MEGA (Adam) and the strongest derivative-free competitor for the linear projection sphere (top), arm repertoire (middle), and latent space illumination (bottom) domains.

resolution of the archive, we normalize QD-score by the archive size (the total number of cells from the tessellation of the measure space). As an additional metric of diversity we compute the coverage as the number of occupied cells in the archive divided by the total number of cells. We run each algorithm for 20 trials in the linear projection and arm repertoire domains, and for 5 trials in the LSI domain, resulting in a total of 445 trials.

## 6.2 Analysis

Table 1 shows the metrics of all the algorithms, averaged over 20 trials for the benchmark domains and over 5 trials for the LSI domain. We conducted a two-way ANOVA to examine the effect of algorithm and domain (linear projection (sphere), linear projection (Rastrigin), arm repertoire) on the QD-Score. There was a statistically significant interaction between the search algorithm and the domain ($F(14, 456) = 7328.18, p < 0.001$). Simple main effects analysis with Bonferroni corrections showed that CMA-MEGA and OMG-MEGA performed significantly better than each of the baselines in the sphere and Rastrigin domains, with CMA-MEGA significantly outperforming OMG-MEGA. CMA-MEGA also outperformed all the other algorithms in the arm repertoire domain.

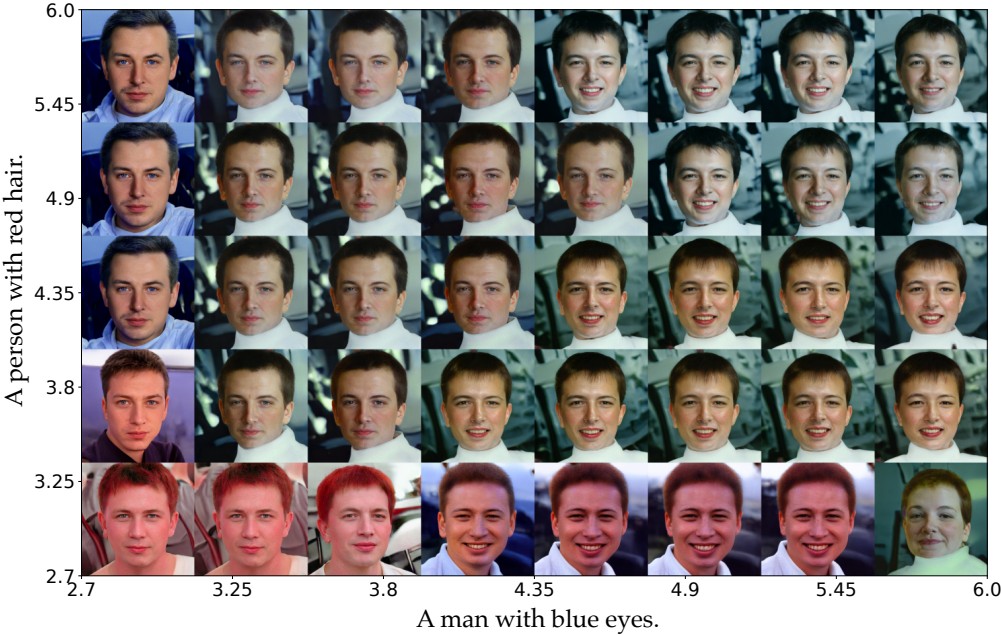

Figure 3: Result of latent space illumination for the objective prompt "Elon Musk with short hair." and for the measure prompts "A person with red hair." and "A man with blue eyes.". The axes values indicate the score returned by the CLIP model, where lower score indicates a better match.

We additionally conducted a one-way ANOVA to examine the effect of algorithm on the LSI domain. There was a statistically significant difference between groups ($F(4, 20) = 260.64, p < 0.001$). Post-hoc pairwise comparisons with Bonferroni corrections showed that CMA-MEGA (Adam) significantly outperformed all other algorithms, while CMA-MEGA without the Adam implementation had the worst performance.

Both OMG-MEGA and CMA-MEGA variants perform well in the linear projection domain, where the objective and measure functions are additively separable, and the partial derivatives with respect to each parameter independently capture the steepest change of each function. We observe that OG-MAP-Elites performs poorly in this domain. Analysis shows that the algorithm finds a nearly perfect best solution for the sphere objective, but it interleaves following the gradient of the objective with exploring the archive as in standard MAP-Elites, resulting in smaller coverage of the archive.

In the arm domain, OMG-MEGA manages to reach the extremes of the measure space, but the algorithm fails to fill in nearby cells. The OG-MAP-Elites variants perform significantly better than OMG-MEGA, because the top-performing solutions in this domain tend to be concentrated in an "elite hypervolume" [61]; moving towards the gradient of the objective finds top-performing cells, while applying isotropic perturbations to these cells fills in nearby regions in the archive. CMA-MEGA variants retain the best performance in this domain. Fig. 1 shows a high-precision view of the CMA-MEGA (Adam) archive for the arm repertoire domain.

We did not observe a large difference between the CMA-MEGA (Adam) and our gradient descent implementation in the first two benchmark domains, where the curvature of the search space is well-conditioned. On the other hand, in the LSI domain CMA-MEGA without the Adam implementation performed poorly. We conjecture that this is caused by the conditioning of the mapping from the latent space of the StyleGAN to the CLIP score.

Fig. 2 shows the QD-score values for increasing number of iterations for each of the tested algorithms, with 95% confidence intervals. The figure also presents heatmaps of the CMA-MEGA (Adam) and the generated archive of the strongest QD competitor for each of the three domains. We provide generated archives of all algorithms in the supplemental material.

We visualize the top performing solutions in the LSI domain by uniformly sampling solutions from the archive of CMA-MEGA (Adam) and showing the generated faces in Fig. 3. We observe that

as we move from the top right to the bottom left, the features matching the captions "a man with blue eyes" and "a person with red hair" become more prevalent. We note that these solutions were generated from a single run of CMA-MEGA (Adam) for 10,000 iterations.

Overall, these results show that using the gradient information in quality diversity optimization results in significant benefits to search efficiency, but adapting the gradient coefficients with CMA-ES is critical in achieving top performance.

# 7 Related Work

**Quality Diversity**. The precursor to QD algorithms [50] originated with diversity-driven algorithms as a branch of evolutionary computation. Novelty search [39], which maintains an archive of diverse solutions, ensures diversity though a provided metric function and was the first diversity-driven algorithm. Later, objectives were introduced as a quality metric resulting in the first QD algorithms: Novelty Search with Local Competition (NSLC) [40] and MAP-Elites [13, 43]. Since their inception, many works have improved the archives [18, 62, 57], the variation operators [61, 19, 11, 46], and the selection mechanisms [12, 56] of both NSLC and MAP-Elites. While the original QD algorithms were based on genetic algorithms, algorithms based on other derivative-free approaches such as evolution strategies [19, 10, 46, 11] and Bayesian optimization [37] have recently emerged.

Being stochastic derivative-free optimizers [7], QD algorithms are frequently applied to reinforcement learning (RL) problems [47, 2, 14] as derivative information must be estimated in RL. Naturally, approaches combining QD and RL have started to emerge [45, 9]. Unlike DQD, these approaches *estimate* the gradient of the reward function, and in the case of QD-RL a novelty function, in action space and backpropagate this gradient through a neural network. Our proposed DQD problem differs by leveraging *provided* – rather than approximated – gradients of the objective and measure functions.

Several works have proposed model-based QD algorithms. For example, the DDE-Elites algorithm [23] dynamically trains a variational autoencoder (VAE) on the MAP-Elites archive, then leverages the latent space of this VAE by interpolating between archive solutions in latent space as a type of crossover operator. DDE-Elites learns a manifold of the archive data as a representation, regularized by the VAE's loss function, to solve downstream optimization tasks efficiently in this learned representation. The PoMS algorithm [52] builds upon DDE-Elites by learning an *explicit* manifold of the archive data via an autoencoder. To overcome distortions introduced by an explicit manifold mapping, the authors introduce a covariance perturbation operator based on the Jacobian of the decoder network. These works differ from DQD by dynamically constructing a learned representation of the search space instead of leveraging the objective and measure gradients directly.

**Latent Space Exploration**. Several works have proposed a variety of methods for directly exploring the latent space of generative models. Methods on GANs include interpolation [60], gradient descent [3], importance sampling [64], and latent space walks [33]. Derivative-free optimization methods mostly consist of latent variable evolution (LVE) [4, 24], the method of optimizing latent space with an evolutionary algorithm. LVE was later applied to generating Mario levels [63] with targeted gameplay characteristics. Later work [20] proposed latent space illumination (LSI), the problem of exploring the latent space of a generative model with a QD algorithm. The method has only been applied to procedurally generating video game levels [20, 58, 55, 17] and generating MNIST digits [65]. Follow-up work explored LSI on VAEs [54]. Our work improves LSI on domains where gradient information on the objective and measures is available with respect to model output.

# 8 Societal Impacts

By proposing gradient-based analogs to derivative-free QD methods, we hope to expand the potential applications of QD research and bring the ideas of the growing QD community to a wider machine learning audience. We are excited about future mixing of ideas between QD, generative modeling, and other machine learning subfields.

In the same way that gradient descent is used to synthesize super-resolution images [42], our method can be used in the same context, which would raise ethical considerations due to potential biases present in the trained model [6]. On the other hand, we hypothesize that thoughtful selection of the measure functions may help counterbalance this issue, since we can explicitly specify the measures

that ensure diversity over the collection of generated outputs. For example, a model may be *capable* of generating a certain type of face, but the latent space may be *organized* in a way which biases a gradient descent on the latent space away from a specific distribution of faces. If the kind of diversity required is differentiably measurable, then DQD could potentially help resolve which aspect of the generative model, i.e., the structure of the latent space or the representational capabilities of the model, is contributing to the bias.

Finally, we recognize the possibility of using this technology for malicious purposes, including generation of fake images ("DeepFakes"), and we highlight the utility of studies that help identify DeepFake models [28].

## 9 Limitations and Future Work

Quality diversity (QD) is a rapidly emerging field [7] with applications including procedural content generation [27], damage recovery in robotics [13, 43], efficient aerodynamic shape design [22], and scenario generation in human-robot interaction [16, 17]. We have introduced differentiable quality diversity (DQD), a special case of QD, where measure and objective functions are differentiable, and have shown how a gradient arborescence results in significant improvements in search efficiency.

As both MEGA variants are only first order differentiable optimizers, we expect them to have difficulty on highly ill-conditioned optimization problems. CMA-ES, as an approximate second order optimizer, retains a full-rank covariance matrix that approximates curvature information and is known to outperform quasi-Newton methods on highly ill-conditioned problems [25]. CMA-ME likely inherits these properties by leveraging the CMA-ES adaptation mechanisms and we expect it to have an advantage on ill-conditioned objective and measure functions.

While we found CMA-MEGA to be fairly robust to hyperparameter changes in the first two benchmark domains (linear projection, arm repertoire), small changes of the hyperparameters in the LSI domain led CMA-MEGA, as well as all the QD baselines, to stray too far from the mean of the latent space, which resulted in many artifacts and unrealistic images. One way to address this limitation is to constrain the search region to a hypersphere of radius $\sqrt{d}$, where $d$ is the dimensionality of the latent space, as done in previous work [42].

While CLIP achieves state-of-the-art performance in classifying images based on visual concepts, the model does not measure abstract concepts. Ideally, we would like to specify "age" as a measure function and obtain quantitative estimates of age given an image of a person. We believe that the proposed work on the LSI domain will encourage future research on this topic, which we would in turn be able to integrate with DQD implementations to generate diverse, high quality content.

Many problems, currently modelled as optimization problems, may be fruitfully redefined as QD problems, including the training of deep neural networks. Our belief stems from recent works [53, 41], which reformulated deep learning as a multi-objective optimization problem. However, QD algorithms struggle with high-variance stochastic objectives and measures [35, 15], which naturally conflicts with minibatch training in stochastic gradient descent [5]. These challenges need to be addressed before DQD training of deep neural networks becomes tractable.

## Acknowledgments and Disclosure of Funding

We would like to thank the anonymous reviewers fNzE, 4FYu, T1rp, and ghwT for their detailed feedback, thorough comments, and corrections throughout the review process that helped us improve the quality of the paper. We thank the area chair and senior area chair for their assessment, paper recommendation, and kind words about the paper. We thank Lisa B. Soros for her feedback on a preliminary version of this work and Varun Bhatt for his assistance running additional experiments for the final version of the paper.

This work was partially supported by the National Science Foundation NRI (# 2024936) and the Alpha Foundation (# AFC820-68).

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
