# OpenReview forum: "Differentiable Quality Diversity"
_NeurIPS.cc/2021/Conference — NeurIPS 2021 Oral_

### Official Review · Reviewer_ghwT · 2021-07-13

**Rating:** 8
**Confidence:** 5

**Summary:**

In the paper "Differentiable Quality-Diversity", the authors propose a new set of algorithms quality diversity algorithms for differentiable problems. Quality-Diversity is a new family of stochastic optimisation algorithms, often applied to black-box optimisation problems, where the goal is to generate a large collection of diverse and high-performing solutions. This paper provides novel approaches to leverage gradient information when the problems considered are first-order differentiable. The proposed methods (CMA-MEGA and OMG-MEGA) demonstrate excellent performance compared to state-of-the-art Quality-Diversity algorithms.


**Limitations And Societal Impact:**

I cannot identify potential direct societal impacts of their work.

**Main Review:**

The paper is well written and clearly structured. It is pleasant to reach and easy to understand without unnecessary technical definitions and notations.
The proposed methods make a lot of sense and are a smart way of using the gradient information when available. In particular, the combination of the measure and fitness value in a linear combination which then uses CMA-ES to automatically adjust the weights depending on the state of the archive is a nice way to formalise the problem. The use of the proposed methods in the domain of latent space illumination demonstrates promising results.

The introduction and literature review about quality diversity is appropriate and is likely understandable by non-expert readers.

Here are a couple of mostly minor points and inaccuracies:

1)It might be important to stress that the proposed methods and associated performance improvements are limited to a specific scope of application (first-order differentiable) and that not all problems fall in this category. For instance, the paper says "we focus on domains with differentiable objective and measures, where DQD is applicable.". This is not an intentional choice but imposed by the method.

2) "MAP-Elites first samples solutions from a fixed distribution θ ∼ N (0, I )". This is not always the case. Multiple papers (in particular from MAP-Elites' authors, including the reference implementation) use a uniform sampling (which require the search space to be bounded). This is a minor detail but can have a significant impact on certain tasks, like the linear projection considered in this paper (a 0 centred normal distribution is likely to make the problem easier).

3) It is correct to say that PGA-ME combines a gaussian variation operator (the Iso+LineDD operator to be exact) with a policy gradient operator only on the objective. However, PGA-ME does not sum (or successively apply) the two operators on the same individual. Instead, it randomly applies one of those for each new offspring. The OG-ME variant introduced in this paper sums these two operators which is likely to create mutations that are too large to be effective.

4) I did not find the definition of the weights wi used in line 14 of algorithm 1.

5) I understand the desire to make some benchmarks more challenging, but I think that a 1000dof planar robot is a bit too much in my opinion (in particular when considering that this benchmark was initially based on a real 8dof robot). I hope that in the future the community will come up with realistic domains that will be challenging enough to demonstrate the benefits of such methods.

6) The value of theta_0 used in the experiments cannot be found in the paper and supplementary material.

7)Table 1 reports a surprising result: CMA-ME in the two LP domains seems to perform like MAP-Elites and worse than MAP-Elites (line), which goes against the main result of the original CMA-ME paper (which uses the same domains). The same thing is observed in the table provided in the supplementary material. However, I cannot find a discussion about this in the paper.

8)In the sentence "To make the QD-score invariant with respect to the resolution of the archive, we normalize QD-score by the archive size", does "archive size" refer to the maximal archive size, or the number of solutions in the archive? I believe it is the former, but this could be clarified to avoid confusion.

9) In "Naturally, approaches combining QD and RL have started to emerge [42, 9]. Unlike DQD, these approaches estimate the gradient of the reward function in action space and backpropagate this gradient through a neural network. Our proposed DQD problem differs by leveraging provided gradients for both the objective and measure functions.", it is important to note that QD-RL [9] introduces a diversity gradient (via a diversity critic), and uses both the policy gradient (reward-based) and the diversity gradient. Therefore, it is inaccurate to say that DQD differs from other approaches because it uses gradient at both levels. Additionally, QD-RL is not limited to differentiable problems.


**Time Spent Reviewing:**

5

---

> ### Author Response · Authors · 2021-08-10
> **Response to Reviewer ghwT**
>
> **It might be important to stress that the proposed methods and associated performance improvements are limited to a specific scope of application (first-order differentiable) and that not all problems fall in this category. For instance, the paper says "we focus on domains with differentiable objective and measures, where DQD is applicable.". This is not an intentional choice but imposed by the method.**
>
> We agree with the reviewer.  We will clarify that the DQD algorithm is designed to work in settings where the objective and measures are differentiable. We hypothesize that DQD algorithms can be also applied to derivative-free QD problems by approximating gradients via finite difference methods and we will investigate this in future work.
>
> **"MAP-Elites first samples solutions from a fixed distribution θ ∼ N (0, I )". This is not always the case. Multiple papers (in particular from MAP-Elites' authors, including the reference implementation) use a uniform sampling (which require the search space to be bounded). This is a minor detail but can have a significant impact on certain tasks, like the linear projection considered in this paper (a 0 centred normal distribution is likely to make the problem easier).**
>
> In the LSI domain we let the initial distribution in MAP-Elites to be a Gaussian because the GAN was trained on a latent space sampled from a fixed variance Gaussian distribution. We then set the initial distribution to be Gaussian in all the experiments for consistency.
>
> We agree with the reviewer that the initial distribution of solutions in MAP-Elites does not need to be Gaussian. We note that switching to uniform sampling would not affect our results in the linear projection domain. Specifically in the linear projection domain, the measure space is formed by summing the parameters which are random variables. As the number of random variables increases, the distribution of their sum becomes normal because of the central limit theorem. Since we use a large number of parameters (n=1000), we expect the distribution in measure space of an initial population of uniformly sampled solutions to approximate a Gaussian, similarly to having a Gaussian initial population. To verify that the initial distribution would not affect the results in our experiments, we reran the LP and arm experiments by changing the term np.random.normal(size=dim) on line 259 of lin_proj.py in the code included with the supplemental material to np.random.uniform(low=-1, high=1, size=dim). The results were within variance of our presented results.
>
> **It is correct to say that PGA-ME combines a gaussian variation operator (the Iso+LineDD operator to be exact) with a policy gradient operator only on the objective. However, PGA-ME does not sum (or successively apply) the two operators on the same individual. Instead, it randomly applies one of those for each new offspring. The OG-ME variant introduced in this paper sums these two operators which is likely to create mutations that are too large to be effective.**
>
> We agree with the reviewer about the difference between OG-ME and PGA-ME, which we will emphasize in the revised version. Overall, we selected the OG-ME baseline in order to demonstrate the importance of gradients for exploring the measure space, compared to using Gaussian perturbations. We note that in OG-ME we evaluate the intermediate solution after perturbing the solution, add it to the archive (if it improves the archive), take a gradient step, re-evaluate and then add to the archive. This means that solutions are generated via alternating between perturbation operations and gradient steps. Our motivation was that, since we need to evaluate a solution to compute its gradient, for the same number of total evaluations we can take more gradient steps overall, if we apply a gradient step after each solution has been evaluated once.
>
> To test the performance if we apply the two operators to different offsprings, we reran the experiments with two separate operators (one gradient descent, one Iso+LineDD). The results were comparable to ME (line) on the LP domain and to the current OG-ME on the arm domain. The reviewer can replicate these results by changing the “emitters” list for OG-MAP-Elites in the provided code to have one GradientEmitter and one IsoLineEmitter emitters instead of just one emitter and changing the sigma0 parameter in the GradientEmitter to 0.0.
>
> **I did not find the definition of the weights wi used in line 14 of algorithm 1.**
>
> w_i refers to the ranking weights from the CMA-ES algorithm (see [28]). We refer to w_i on line 145 and we will update this description to explicitly mention w_i for improved clarity.
>
> **I understand the desire to make some benchmarks more challenging, but I think that a 1000dof planar robot is a bit too much in my opinion (in particular when considering that this benchmark was initially based on a real 8dof robot). I hope that in the future the community will come up with realistic domains that will be challenging enough to demonstrate the benefits of such methods.**
>
> We agree with the reviewer about the need for complex and realistic benchmarks. Our goal in this paper was to demonstrate DQD on domains where QD was shown to work well.  We specifically selected the arm domain, because we can efficiently compute derivatives with inverse kinematics equations and we wanted to show that existing approaches do not take advantage of those derivatives. We anticipate that DQD algorithms will make quality diversity more compatible with modern deep learning methods and we expect many exciting DQD applications to follow from this work.
>
> **The value of theta_0 used in the experiments cannot be found in the paper and supplementary material.**
>
> In all domains we set theta_0 to the zero vector. We will add this information to Appendix A that is included as supplemental material.
>
> **Table 1 reports a surprising result: CMA-ME in the two LP domains seems to perform like MAP-Elites and worse than MAP-Elites (line), which goes against the main result of the original CMA-ME paper (which uses the same domains). The same thing is observed in the table provided in the supplementary material. However, I cannot find a discussion about this in the paper.**
>
> We note that the original CMA-ME paper [18] ran the LP domain for n=20 and n=100, while  here  we have n=1000. The LP domain’s distortions become worse as n increases and that makes the domain much harder to explore. At the same time, CMA-ME must adapt an n^2 = 1 million parameter covariance matrix, which adapts with a slower learning rate than the n=100 case (see [28]). The result is that CMA-ME-generated solutions fail to land in different cells and restarts become more frequent. The large frequency of restarts causes CMA-ME to behave similarly to MAP-Elites. We will clarify this in the revised version.
>
> On the other hand, in lower dimensions, ME (line) performs worse than CMA-ME because ME (line) does not shrink its sampling variance as solutions approach the extremes, which is essential in the LP domain (see discussion in section B in the Appendix that is included as supplemental material).
>
> **In the sentence "To make the QD-score invariant with respect to the resolution of the archive, we normalize QD-score by the archive size", does "archive size" refer to the maximal archive size, or the number of solutions in the archive? I believe it is the former, but this could be clarified to avoid confusion.**
>
> The reviewer is correct, the former is the right interpretation. We will update the paper to clarify this.
>
> **In "Naturally, approaches combining QD and RL have started to emerge [42, 9]. Unlike DQD, these approaches estimate the gradient of the reward function in action space and backpropagate this gradient through a neural network. Our proposed DQD problem differs by leveraging provided gradients for both the objective and measure functions.", it is important to note that QD-RL [9] introduces a diversity gradient (via a diversity critic), and uses both the policy gradient (reward-based) and the diversity gradient. Therefore, it is inaccurate to say that DQD differs from other approaches because it uses gradient at both levels. Additionally, QD-RL is not limited to differentiable problems.**
>
> We agree with the reviewer and we will update the reference to QD-RL. We note that, while QD-RL does not require provided gradients, it is restricted to reinforcement learning problems and is not a general purpose QD optimizer. Additionally, QD-RL is different in that it approximates a diversity gradient of a novelty function (based on novelty search) in action space, rather than using an analytic gradient of the objective and measure functions. The gradient of the novelty function takes the discrete archive as input and it needs to be approximated due to the discrete nature of the novelty search archive.
>
> **Limitations And Societal Impact**
> We refer the reviewer to Appendix F (in the supplemental material) for this discussion. We will move the discussion to the main text.

---

> > ### Comment · Reviewer_ghwT · 2021-08-23
> > **Thank you for your response**
> >
> > Thank you for your response.
> >
> > I am very happy with the changes proposed by the authors, mainly in terms of additional discussion.
> >
> > A few minor comments:
> >
> > - using np.random.uniform(low=-1, high=1, size=dim) is expected to not change the results as of the generated solutions (i.e., 100%) will be within the fold containing all the elites, like the isotropic Gaussian. What happens if np.random.uniform(low=-50, high=50, size=dim) is used?
> >
> > - I appreciate the motivation of OG-ME and the implementation reasons make sense. Yet, from what I understand, using independent operators seems to significantly improve the results (in LP), which is good to know and add to the paper.
> >
> > - The discussion between CMA-ME and ME(line) is very interesting! I did not consider the requirement of inverting the 1000x1000 matrix. I think it would be a nice addition to the paper, for instance in the appendix, for interested readers.
> >
> > - Regarding the societal impact, I meant that I do not think this paper may raise any concern about societal impact. Yet, I am glad to see that a discussion is present in the appendix.
> >
> > Based on the response to my review, I am happy to increase my rating from 7 to 8.

---

> > > ### Author Response · Authors · 2021-08-28
> > > **Thank You and Minor Comments Response**
> > >
> > > Thank you very much for the increase of score and the very thorough feedback. We address each of the additional points below.
> > >
> > > **using np.random.uniform(low=-1, high=1, size=dim) is expected to not change the results as of the generated solutions (i.e., 100%) will be within the fold containing all the elites, like the isotropic Gaussian. What happens if np.random.uniform(low=-50, high=50, size=dim) is used?**
> > >
> > > We ran additional experiments with np.random.uniform(low=-50, high=50, size=dim) with all algorithms that sample initial populations: MAP-Elites, MAP-Elites (line), OG-MAP-Elites, OMG-MEGA.
> > >
> > > | Algorithm           | Sphere Coverage | Rastrigin Coverage |
> > > | --------------------| ----------------|--------------------|
> > > | MAP-Elites          | 1.077%          |0.855%		    |
> > > | MAP-Elites (line)   | 0.922%          |0.812%               |
> > > | OG-MAP-Elites       | 1.581%          |0.795%              |
> > > | OMG-MEGA            | 88.231%         |35.829%             |
> > >
> > > We observe this initial population results in a decrease in performance of all algorithms. The reason is that in the linear projection domain, the measure contribution of each component $x_i$ of a sample $x$ is bounded by a clip function (see Eq. 3 and Fig. 1(b) in the Appendix); if $|x_i| > 5.12$, we set clip(x_i) = 5.12 / x_i, which quickly brings the measure contribution of x_i towards 0. Thus, most samples of the initial population end up concentrated near the center of the measure space and all samples are outside the linear region of the measure space with high probability.
> > >
> > > The sharpest decline was in MAP-Elites (line). As all of the initial solutions found were outside the linear region, we speculate that the performance drop is due to linear interpolation being less effective outside the linear region.
> > >
> > > **I appreciate the motivation of OG-ME and the implementation reasons make sense. Yet, from what I understand, using independent operators seems to significantly improve the results (in LP), which is good to know and add to the paper.**
> > >
> > > We would like to clarify that two aspects of OG-MAP-Elites changed when we ran the above experiment. We changed the alternating operators to independent operators and changed the isotropic Gaussian operator to the Iso+LineDD operator. This week we ran a full ablation for all four combinations to see the effects of each combination of changes.
> > >
> > > We found that alternating between the two operators versus having independent operators doesn’t have a large effect. Alternating yields slightly better results in QD-Score and coverage on the LP domains when you control for the perturbation operator (isotropic Gaussian versus Iso+LineDD). For the arm domain having separate operators, rather than alternating, performs slightly better. In the arm domain, Iso+LineDD and the isotropic Gaussian operator perform similarly.
> > >
> > > The type of perturbation operator is indeed an important consideration and we thank the reviewer for pointing out this difference between PGA-ME and OG-MAP-Elites. We will add an additional baseline called OG-MAP-Elites (line) to the paper that is more consistent with PGA-ME. We will discuss the implementation choices and tradeoffs involving independent versus alternating operators in detail in the appendix.
> > >
> > > **The discussion between CMA-ME and ME(line) is very interesting! I did not consider the requirement of inverting the 1000x1000 matrix. I think it would be a nice addition to the paper, for instance in the appendix, for interested readers.**
> > >
> > > We will additionally update the appendix with a discussion about differences between CMA-ME and ME(line) for large dimensional domains and we thank the reviewer for the suggestion.

---

### Official Review · Reviewer_T1rp · 2021-07-16

**Rating:** 8
**Confidence:** 2

**Summary:**

The authors propose a quality diversity framework whether the objective and measure functions are differentiable. They propose two algorithms to this end by augment existing QD algorithms with gradient information and present experimental results on two benchmarks and a latent space illumination task.

**Limitations And Societal Impact:**

Yes

**Main Review:**

**Update: I've revised my score from 7 to 8, following the author response.**

**Pros**
- The idea of DQD proposed by the paper is an intuitive and natural development combining ideas from QD with modern ML driven by first-order optimization.
- The paper is extremely well-written, with a brief but useful explanation of the background, and the algorithms are carefully described in detail.
- The DQD benchmark with StyleGAN-CLIP is quite intuitive and is bound to inspire some follow-up work.
- The experimental evaluation is convincing, with the proposed algorithms demonstrating clear wins in high dimensional parameter spaces.
- The experimental results are presented with a statistical significance analysis -- I commend the authors for this.

**Cons/Questions**
- My first instinct upon looking at the problem would be to write down an optimization problem which trades-off the objective for diversity, e.g.,maintain a catalog $\theta_1, \cdots, \theta_N$ and iteratively update $\theta_j$ with an approximate solution to $\max_\theta f(\theta) + \lambda \sum_{i \neq j} \Vert m(\theta) - m(\theta_i)\Vert^2$ where $\lambda$ is a trade-off parameter. How is the proposed approach better than this baseline?
- MEGA takes one gradient step in Eq. (3). What are the pros and cons of taking multiple gradient steps? Would this help experimentally?
- The proposed strategy to tune the learning rate in Appendix A does not seem scalable. Could the authors suggest some heuristics to choose the learning rate? How would a grid search work in this case?
- How do you choose the noise variance $\sigma_g^2$? What happens if it is too large or too small?
- It would be helpful if the authors could show plots with QD score and coverage percentage versus the learning rate and $\sigma_g^2$
- (minor) $w_i$ is undefined in Algorithm 1.

**Time Spent Reviewing:**

7 hours

---

> ### Author Response · Authors · 2021-08-10
> **Response to Reviewer T1rp**
>
> **My first instinct upon looking at the problem would be to write down an optimization problem which trades-off the objective for diversity, e.g.,maintain a catalog θ1,⋯,θN and iteratively update θj with an approximate solution to maxθf(θ)+λ∑i≠j∥m(θ)−m(θi)∥2 where λ is a trade-off parameter. How is the proposed approach better than this baseline?**
>
> QD algorithms largely fall into two main families: the Novelty Search with Local Competition (NSLC) family (see [37]) and the MAP-Elites family. The baseline described by the reviewer is in fact similar to the NSLC approach to quality diversity and very similar to a baseline in the NSLC paper. We first compare the baseline suggested by the reviewer with the NSLC approach. We then discuss the strengths and limitations of  the NSLC approach, as compared with the MAP-Elites family. We refer the reviewer to [37] and [11] for a more extended discussion.
>
> The baseline proposed by the reviewer differs from the NSLC approach in the following ways:
>
> (1) Instead of local competition, the baseline encourages global competition. If the global objective term dominates, then the algorithm will overly explore the area close to the nearest local optimum and not fill the measure space. If the measure term dominates, then the search will move to a single extreme point in measure space or fall at a boundary between several cells in the archive and get stuck. On the other hand, local competition scales the objective by the k nearest neighbors in the archive, where nearest is defined by distance in measure space.
>
> (2) The measure term’s computation in the baseline requires going through the whole archive, which is inefficient. In NSLC, the authors instead calculate the distance in measure space from the k nearest neighbors only, where k is a hyperparameter.
>
> (3) NSLC maintains a population of pareto optimal tradeoffs between a local competition objective and a novelty objective defined as the sum of distances from the k nearest neighbors. This approach is an alternative to the trade-off parameter \lambda. The \lambda parameter would need to be selected carefully to avoid domination by one term.
>
> We now briefly discuss how NSLC is different than MAP-Elites. NSLC has the advantage over MAP-Elites in that NSLC dynamically constructs its archive and does not require specified ranges of the measure functions. On the other hand, NSLC ofen falls into cycling patterns, where the search oscillates between two or more areas of measure space.
>
> Finally, we note that maximizing a novelty term (sum of measure function distances)  would push the search towards the extremes. On the other hand, our DQD approach forms a search space where the normalized gradients are the axes and this allows us to find solutions between the extremes. One promising approach could be to take the normalized gradient of the novelty term and use CMA-ES to dynamically adapt the $\lambda$ parameter, so as to take intermediate steps that help fill in the archive. Generally, we expect future work to explore different DQD variants of NSLC and compare those variants to MEGA.
>
> **MEGA takes one gradient step in Eq. (3). What are the pros and cons of taking multiple gradient steps? Would this help experimentally?**
>
> On one hand, by taking one gradient step, we can model the distribution of branching steps as a Gaussian and adapt that Gaussian via CMA-ES in the CMA-MEGA algorithm. On the other hand, taking multiple steps during a branch would allow us to incorporate Adam into the branching computations. Momentum from Adam can improve how branching performs on ill-conditioned measures and objectives. We will discuss this in the revised version and thank the reviewer for pointing this out.
>
> **The proposed strategy to tune the learning rate in Appendix A does not seem scalable. Could the authors suggest some heuristics to choose the learning rate? How would a grid search work in this case?**
>
> Overall, selecting learning rates in MEGA is similar to selecting learning rates in SGD. While we believe that grid search and related methods would work well for tuning MEGA, an alternative and more efficient approach would be to pick a learning rate that is small as possible, but large enough to ensure that branching lands in different cells.The process of finding the smallest learning rate with this property could then be done efficiently via binary search.
>
> **How do you choose the noise variance σg2? What happens if it is too large or too small?**
>
> If σg2 is too large, the gradient steps will be too large. The intuitions about the size of the gradient steps follow those of SGD: if the gradient step is too large, the algorithm will perform similarly to random  search, since it will struggle to follow a local ridge.
>
> **It would be helpful if the authors could show plots with QD score and coverage percentage versus the learning rate and σg2.**
>
> We will include a bar graph in the appendix showing how coverage and QD score change when picking different learning rates and variances for OMG-MEGA and CMA-MEGA. We note that we found the quantitative results to be robust with respect to the hyperparameters. On the other hand, the hyperparameter selection played a role in the realism of the generated images in the LSI domain, because latent codes near the tales of the training distribution often result in unrealistic images. Optimizing the latent space with Adam or RMSProp has similar sensitivities.
>
> **(minor) wi is undefined in Algorithm 1**
>
> w_i refers to the ranking weights from the CMA-ES algorithm (see [28]). We refer to w_i on line 145 and we will update this description to explicitly mention w_i for improved clarity.

---

> > ### Comment · Reviewer_T1rp · 2021-08-24
> > **Revised score**
> >
> > Thank you for your response. I've increased my score from 7 to 8.

---

> > > ### Author Response · Authors · 2021-08-28
> > > **Thank You**
> > >
> > > Thank you very much. We appreciate the increase of the score and the thorough feedback.

---

### Official Review · Reviewer_4FYu · 2021-07-16

**Rating:** 7
**Confidence:** 2

**Summary:**

This work introduces  DQD (differentiable quality diversity) problems: quality-diversity (QD) problems when both the objective function and the measures of diversity are differentiable. The running example for this is obtaining a variety of generated images corresponding to the objective "looks like Elon Musk" and diverse with respect to hair and eye color.

The authors then introduce 2 algorithm variants that can take advantage of differentiability to solve the DQD problem.
- OMG-MEGA uses gradient arborescence to explore based on the diversity measures and optimize with respect to the objective.
- CMA-MEGA branches based on both objective and diversity measures, but optimizes with respect to the objective.
Both proposed algorithms are variants on QD solutions (resp. MAP-Elites and and CMA-ME) that do not assume access to gradient information.

Across several experimental evaluations, the authors show that their proposed algorithms, and especially CMA-MEGA, outperform baselines that do not access gradient information, both in terms of maximizing the QD score and in terms of coverage of the search space.

**Limitations And Societal Impact:**

The limitations and negative societal impact have been discussed in detail.

**Main Review:**

Originality: This work builds upon prior solutions to building archives for the QD problem, taking advantage of use cases in which gradient information is available for both the QD objective and diversity metrics.

Quality: This work reads as a complete paper, with a careful analysis of the pros and cons of the proposed methods (e.g., computational complexity constraints when applied to experimental design settings). Being unfamiliar with the relevant QD literature, I would have found this paper easier to parse were it to include a more in-depth discussion of several key steps of the proposed algorithm. In particular:
- how important is normalizing the gradient in OMG-MEGA and CMA-MEGA?
- can you formalize your claim "$\sigma_g$ acts as a learning rate for the gradient step"?

Clarity: Without familiarity with the QD literature, the distinction between the CMA-MEGA and baseline CMA-ME algorithm is difficult to parse. Algorithm 1 could be augmented to include comments indicating deviations from the CMA-ME algorithm for increased clarity.
I also found the running Elon Musk example somewhat counter-intuitive, and Figure 3 difficult to read (the eye color is not easily visible). I do not know if this is easily feasible, but choosing an example with more phenotype variation (dog breeds?) that is less reliant on small detail variations would have increased readability (in my opinion).

Significance:
The results seem significant to the QD community, as the introduced algorithms, and CMA-MEGA variants in particular, show vastly improved coverage of the search space compared to previous QD baselines.

Minor comments and questions:
- I believe the correct reference for MAP-Elites is [40] as well as [13] (last paragraph, page 1)
- When you mention the QD objective (line 121), are you referring to $f(\theta)$?
- In Figure 3 of the appendix, it seems that adding the Adam variation can make a significant impact on the coverage. Do you have any intuition why this may be the case?


**Time Spent Reviewing:**

3 hours

---

> ### Author Response · Authors · 2021-08-10
> **Response to Reviewer 4FYu**
>
> In the reviewer’s summary, we would like to clarify that CMA-MEGA branches according to the objective and measures and ascends according to the QD objective (see Eq 1), rather than the objective function f.
>
> **How important is normalizing the gradient in OMG-MEGA and CMA-MEGA?**
>
> Normalizing gradients is important for the following reasons:
>
> (1) Suppose that we did not normalize the gradients on the LP (sphere) domain. Note that the objective function is quadratic. As we move further from the objective’s optimum, the magnitude of the gradient of the objective increases. This means that when exploring areas of the search space with low objective value, the objective gradient will start dominating the measure gradients, hindering exploration. The same holds for the LP (Rastrigin) domain, because the Rastrigin objective is bumpy and multi-modal. On the other hand, CMA-MEGA will shrink the objective coefficient. This results in CMA-MEGA obtaining 100% coverage of measure space, even with a bumpy objective.
>
> (2) The objective function and measures could each be on different scales. The function with the largest derivative values would dominate the exploration if we did not normalize.
>
> (3) If we do not normalize the gradient, changing the objective and measures by multiplying each function by a constant would change the behavior of the algorithm. On the other hand, normalizing ensures that the performance is invariant to linear transformations of the objective and measures.
>
> (4) Normalization helps with stability. We note that CMA-ES is solving a non-stationary optimization problem for gradient coefficients that maximize the QD objective. The coefficients can be viewed as a learning rate for each gradient. In this sense, CMA-MEGA has adaptive learning rates where the learning rates are maintained by CMA-ES. If we are both changing the magnitude of the gradients and we have adaptive learning rates, then the two may end up fighting each other. This can cause oscillations in CMA-ES’s adaptation mechanisms and lead to instability.
>
> In addition to the justification above, in the revised version, we will include an additional study with unnormalized gradients.  The reviewer can preview the results of the study by changing the normalize_gradients flag to false during the setup of each algorithm in lin_proj.py and arm.py in code provided as part of the supplementary material.
>
> **Can you formalize your claim "σg” acts as a learning rate for the gradient step"?**
>
> We note  that in standard gradient descent the learning rate is a scalar multiplier for the gradient. In OMG-MEGA, we select a coefficient for the gradient from a Gaussian distribution with standard deviation σg. If we increase "σg” the algorithm will take larger gradient steps on each branch. In contrast, if we decrease “σg” the algorithm will take smaller gradient steps on each branch. We will clarify this in the revised version.
>
>
> **Readability Issues with the LSI example**
>
> We agree with the reviewer that generating images for dogs would improve readability. We note that our pretrained StyleGAN model was trained on the CelebA dataset. Generating images for dogs would require retraining the model on a different data set or exchanging StyleGAN for a different pretrained model, which we consider as beyond the scope of this work. However, for completeness and diversity, we will include two additional runs of LSI with different text for the objective and measures in the appendix of the revised version of the paper. We will include qualitative and quantitative results for each run similar to our current evaluation.
>
> **I believe the correct reference for MAP-Elites is [40] as well as [13] (last paragraph, page 1)**
>
> This is correct, we will fix this in the revised version.
>
> **When you mention the QD objective (line 121), are you referring to f(θ)?**
>
> We are referring to the QD objective in Eq. 1. We will add a reference “(see Eq. 1)” to increase clarity.
>
> **In Figure 3 of the appendix, it seems that adding the Adam variation can make a significant impact on the coverage. Do you have any intuition why this may be the case?**
>
> Adam contains second moment updates, which enables robustness to ill-conditioned objectives. In contrast, vanilla gradient ascent updates can struggle (oscillate) on ill-conditioned objectives. We note that the arm and LP domains have well-conditioned objectives and measures, while the latent space of the StyleGan is ill-conditioned (see “Analyzing and improving the image quality of StyleGAN” by Karras et al.). We hypothesize that the well-conditioned curvature of the LP domains is the reason Adam is not necessary to solve them. We base our hypothesis on the Adam paper [35], which discusses ill-conditioned curvature in the context of Adam, as well as on Roger Grosse’s CSC 421 Lecture 7-8 slides on optimization in his deep learning course at UToronto, which intuitively visualizes these challenges.

---

> > ### Comment · Reviewer_4FYu · 2021-08-26
> > **Revised score**
> >
> > I thank the authors for their detailed answer to my questions and for their clarifications; I will increase my score to 7,

---

> > > ### Author Response · Authors · 2021-08-28
> > > **Thank You**
> > >
> > > Thank you very much for increasing the score and for the detailed comments.

---

### Official Review · Reviewer_fNzE · 2021-07-17

**Rating:** 7
**Confidence:** 3

**Summary:**

The paper makes three primary contributions:

(1) The paper proposes a new problem setup, called Differentiable Quality Diversity (DQD), where the goal is to optimize for a set of diverse solutions that maximize a target objective. Diversity is formalized by assuming the existence of **differentiable** measure functions, which intuitively specify an axis of variation and thus far away solutions along in the joint space of these measure functions are more diverse.

(2) To solve the DQD, the paper extends the MAP-Elite algorithms from the related and previously studied non-differentiable variant, abbreviated as simply QD. The extensions incorporate gradient information in novel ways, interfacing first-order methods with evolutionary search style procedures.

(3) Finally, the authors conduct empirical evaluation of the proposed algorithms which suggests that these combinations of gradient and evolutionary algorithms are indeed better than either approach alone on some toy benchmark tasks and a generative modeling task.

**Ethical Concerns:**

The paper uses a generative modeling task as an application. Like any other application, the model could have biases but with human faces as the domain of interest, the societal and ethical concerns are even more barebones. The paper has acknowledged these concerns to a reasonable degree.

**Limitations And Societal Impact:**

Yes, the limitations are well described. I would encourage the authors to move the societal impact section from the appendix to the main text, especially given the rising concerns and useful applications of generative modeling. Also, they should include examples of LSI covering diverse demographics and not restrict to Elon Musk.

**Main Review:**

To the best of my knowledge, both the problem setup and the proposed solutions are novel. The setup itself is a non-trivial important special case of the previously studied Quality Diversity problem, eg, the use case in generative modeling has potential implications beyond just diverse face generation in other scientific fields as well. The methods proposed in the work are sound, simple, and easy to implement. I also liked their descriptions, which was sufficiently clear in text and aptly formalized mathematically.

Questions/comments for clarification and improvement:

- In CMA-MEGA, could the authors study the effect of the ranking strategy on downstream performance? in particular, it is not obvious to me that prioritizing all exploration candidates over  the delta improvements is a good defacto strategy. If the delta improvement is very large, then deprioritizing it to explore a new cell is not ideal. Similar to explore-exploit strategies in bayesopt, RL algorithms, an acquisition function style approach to scalarize the exploration score and the delta improvement score seems like a better strategy.

- In general, for the QD problem, we expect a Pareto optimal frontier of (max or top-m) QD score vs. coverage. The results in Table 1 focus on a specific point on this frontier. Can we tailor the algorithms to target a specific frontier? (i.e., for x% coverage, what is the best QD score we can obtain? Or vice versa)

- Empirical trends of model performance as a function of number of cells would be interesting to include for testing the scalability of the approaches and also for monitoring the divergence in performance from a baseline gradient-only approach (at the extreme, when number of cells is 1, we only care about the target based objective and the gradient based approach is expected to dominate).

- Latent space illumination is a very important problem and a very convincing motivation for this work. But the empirical investigations seemed lacking. While the authors acknowledge that the results are not very robust to hyperparameters, a further investigation as to analyzing the causes of instability and tricks to alleviate them would make the work more extendable by other practitioners. The current set of results based on 1 elon musk prompt are too limited and potentially subject to cherrypicking.





**Time Spent Reviewing:**

5

---

> ### Author Response · Authors · 2021-08-10
> **Response to Reviewer fNzE**
>
> **In CMA-MEGA, could the authors study the effect of the ranking strategy on downstream performance? In particular, it is not obvious to me that prioritizing all exploration candidates over the delta improvements is a good defacto strategy. If the delta improvement is very large, then deprioritizing it to explore a new cell is not ideal. Similar to explore-exploit strategies in bayesopt, RL algorithms, an acquisition function style approach to scalarize the exploration score and the delta improvement score seems like a better strategy.**
>
> We note that in CMA-MEGA we adopted the improvement ranking method proposed in the CMA-ME paper [18]. Specifically, the CMA-ME paper proposed three ranking methods: optimizing, improvement, and a ranking based on projective geometry. The authors found the improvement ranking to offer the best balance between optimizing existing solutions and exploring new solutions. While we provide the intuition of CMA-ME’s improvement ranking method below, we note that future work may indeed discover better rankings than CMA-ME’s improvement ranking and such an innovation would in fact benefit both CMA-ME and CMA-MEGA.
>
> First, the two-stage improvement ranking that prioritizes exploration candidates (newly discovered cells) makes the algorithm invariant when a constant is added to the objective function.  Invariances on algorithmic behavior are in general desirable properties in evolution strategies. Especially given that our optimization problem is non-stationary, robustness is essential in discovering good branching coefficients (see discussion in Appendix D included in the supplemental material). Suppose that instead of the current two-stage ranking, we sorted candidate solutions by their objective value if we discovered a new cell, or the improvement delta if we improved upon an existing cell. Ranking in this way would mean that the performance/behavior of the algorithm would change if a constant were added to the objective function, losing the invariance property.
>
>
> Second, we hypothesize that ranking candidate solutions with the alternative way proposed above would cause oscillations between the edge of the frontier of new cells and the existing high-performing cells: Assume that CMA-ME is exploring the frontier of an area of discovered cells, but there is a region of high-performing cells inside that area. Everytime CMA-ME significantly improved a solution in the region of high-performing cells, it would leave the frontier and reverse back to that region, which would slow down exploration of the archive. On the other hand, the two-stage ranking prioritizes coverage of the archive first. Once the archive is saturated, it then focuses on maximizing the objective in each cell.
>
> We agree with the reviewer that adopting explore-exploit strategies is a very interesting direction for future work. We note that explore-exploit strategies often struggle with non-stationary objectives and in DQD the QD optimization problem changes when the archive is updated and new gradients are computed. Thus, we expect that acquisition function approaches that assume a stationary objective would be incompatible with our approach.
>
> **In general, for the QD problem, we expect a Pareto optimal frontier of (max or top-m) QD score vs. coverage. The results in Table 1 focus on a specific point on this frontier. Can we tailor the algorithms to target a specific frontier? (i.e., for x% coverage, what is the best QD score we can obtain? Or vice versa)**
>
> We note that QD score and coverage are non-competing objectives and the Pareto front consists of just one solution. To prove this, consider two different archives on the Pareto front such that one archive has better QD score than coverage and the other has better coverage than QD score. A better archive can be formed by unioning the archive and retaining the best solution for each cell between the two archives. This contradicts that the two archives were on the Pareto front.
>
> Although the QD score and coverage are non-competing objectives, we note that the suboptimal archives returned by each search algorithm may indeed exhibit a tradeoff between performance of cells and coverage. The QD-score is an aggregate measure and does not inform us about this tradeoff, but Fig. 4 of the Appendix is more informative: it shows the minimum objective value of any top x% of cells in the archive for each domain and search algorithm. Given a top x% of cells (y-axis), the curve shows us the threshold (x-axis) from which the top x% have greater than (or equal) objective value.
>
> **Empirical trends of model performance as a function of number of cells would be interesting to include for testing the scalability of the approaches and also for monitoring the divergence in performance from a baseline gradient-only approach (at the extreme, when number of cells is 1, we only care about the target based objective and the gradient based approach is expected to dominate).**
>
> We agree with the reviewer about the importance of the number of cells in the archive. In fact, exploring the effect of archive resolution in the MAP-Elites family of QD algorithms is still an open problem and an active area of research in the QD community. One approach [17] is to scale the resolution of the archive over time during the search, based on how solutions are distributed. We are excited about future work on DQD that will explore similar properties.
>
> **Latent space illumination is a very important problem and a very convincing motivation for this work. But the empirical investigations seemed lacking. While the authors acknowledge that the results are not very robust to hyperparameters, a further investigation as to analyzing the causes of instability and tricks to alleviate them would make the work more extendable by other practitioners. The current set of results based on 1 elon musk prompt are too limited and potentially subject to cherrypicking. [...] Also, they should include examples of LSI covering diverse demographics and not restrict to Elon Musk.**
>
> We would like to clarify that the sensitivity to hyperparameters affects the qualitative results, but not the quantitative results. This sensitivity is largely due to how GANs are trained (sampling from a fixed Gaussian) and optimizing the latent space with Adam or RMSProp has similar sensitivities; latent codes near the tales of this Gaussian often result in unrealistic images, so the hyperparameter selection focuses on producing images that look realistic. There are techniques that constrain the search near the “realistic” regions of latent space, such as projecting latent codes on a hypersphere [39] or adding regularization to the objective (see “Compressed sensing of using generative models” by Bora et al.). We expect future work on LSI to exploit such techniques to improve robustness to the hyperparameters.
>
> For completeness and diversity, we will include in the Appendix two additional runs of LSI with different text for the objective and measures. One of the runs will have “Beyonce” as the objective. We will include qualitative and quantitative results for each run similar to our current Elon Musk evaluation.
>
>  **I would encourage the authors to move the societal impact section from the appendix to the main text, especially given the rising concerns and useful applications of generative modeling.**
>
> We agree with the reviewer and will move the societal impact section to the main text.

---

> > ### Comment · Reviewer_fNzE · 2021-08-25
> > **Response to Rebuttal**
> >
> > Thanks for the response. Overall, I remain positive about the paper. I would highly encourage the authors to seriously consider the feedback and stick to their promised enhancements --- I am optimistic it would greatly enrich the readership and long term value of this work!

---

> > > ### Author Response · Authors · 2021-08-28
> > > **Thank You**
> > >
> > > Thank you very much for the thorough review and feedback.

---

### Author Response · Authors · 2021-08-10
**Response to all: Thank you very much for the rigorous reviews.**

Thank you very much for the thorough reviews and insightful comments. We are excited that reviewers appreciated the novelty and significance of the work. We are also very pleased with the recognition that the idea of DQD is “an intuitive and natural development” and that our methods are “​​aptly formalized mathematically” and “make a lot of sense and are a smart way of using the gradient information when available”. We are excited with the remark that the StyleGAN+CLIP benchmark is “bound to inspire some follow-up work”. We are also glad that the reviewers appreciated the statistical analysis of the results and found the paper “extremely well-written” and “pleasant to read and easy to understand.”

We respond to each reviewer separately and look forward to continuing the discussion.

---

### Decision · Program_Chairs · 2021-09-27

**Decision:**

Accept (Oral)

**Comment:**

Meta-review of Differentiable Quality Diversity

This paper proposes the first differentiable version of “Quality Diversity” optimization. QD, along with other multi-objective optimization methods, look at generating a large collection of diverse solutions, and are well-explored in the evolutionary computation community, while lesser known in the ML / RL community.

The 3 main contributions of this paper are well-summarized by reviewer fNzE:

1. The paper proposes a new problem setup, called Differentiable Quality Diversity (DQD), where the goal is to optimize for a set of diverse solutions that maximize a target objective. Diversity is formalized by assuming the existence of differentiable measure functions, which intuitively specify an axis of variation and thus far away solutions along in the joint space of these measure functions are more diverse.

2. To solve the DQD, the paper extends the MAP-Elite algorithms from the related and previously studied non-differentiable variant, abbreviated as simply QD. The extensions incorporate gradient information in novel ways, interfacing first-order methods with evolutionary search style procedures.

3. Finally, the authors conduct empirical evaluation of the proposed algorithms which suggests that these combinations of gradient and evolutionary algorithms are indeed better than either approach alone on some toy benchmark tasks and a generative modeling task.

Usually, the work in QD in the EC community focuses on artificial agents (like simulated robots), but here, the work really sheds light on how QD can be applied and effectively used in applications that the DL community is also interested in, like image generation. As Reviewer 4FYU pointed out: *the running example for this is obtaining a variety of generated images corresponding to the objective "looks like Elon Musk" and diverse with respect to hair and eye color*. Reviewer T1rp agrees, and mentions *The DQD benchmark with StyleGAN-CLIP is quite intuitive and is bound to inspire some follow-up work.*

All in all, the reviewers, including myself, agree that this paper is very well written, with good background explanation, well motivated, and presents experiments that are not only convincing, but will attract many folks in the NeurIPS community especially those in generative modelling to start using QD.

Given the feedback about the generally high quality of the paper and potential impact, I would like to recommend acceptance as Spotlight presentation.